# Cost-Aware Design and Fabrication of New Support Structures in Laser Powder Bed Fusion: Microstructure and Metallurgical Properties

Bharath Bhushan Ravichander [1], Sourabh Thakare [1], Aditya Ganesh-Ram [1], Behzad Farhang [1], Manjunath Hanumantha [1], Yiran Yang [2], Narges Shayesteh Moghaddam [1] and Amirhesam Amerinatanzi [1,3,*]

[1] Mechanical and Aerospace Engineering, University of Texas at Arlington, Arlington, TX 76019, USA; bharathbhushan.ravichander@mavs.uta.edu (B.B.R.); sourabhhemant.thakare@mavs.uta.edu (S.T.); adityakrishna.ganeshram@mavs.uta.edu (A.G.-R.); behzad.farhang@mavs.uta.edu (B.F.); manjunath.hanumantha@mavs.uta.edu (M.H.); narges.shayesteh@uta.edu (N.S.M.)
[2] Industrial, Manufacturing, and Systems Engineering, University of Texas at Arlington, Arlington, TX 76019, USA; yiran.yang@uta.edu
[3] Materials Science and Engineering, University of Texas at Arlington, Arlington, TX 76019, USA
* Correspondence: amir.ameri@uta.edu; Tel.: +1-(817)-272-1286

**Abstract:** This study investigates the effect of support structures on the properties of Inconel 718 (i.e., IN718) parts produced by the laser powder bed fusion (LPBF) additive manufacturing process. Specifically, the effects of support structure shape (i.e., pin-type, angled-type, cone-type) and geometry (i.e., support wall thickness, and gap) on their composition, hardness, microstructure, and material/time consumption are investigated and compared to the conventionally fabricated Inconel 718. From the microstructural analysis, the deepest melt pools appeared to be formed in the sample fabricated on top of the pin-type support structure having a relatively low wall thickness. The XRD results conveyed that a proper selection of geometrical variables for designing support structure results in elevated levels of the strengthening phases of IN718. The sample fabricated on top of the pin-type support structure showed the highest Vickers hardness value of 460.5 HV, which was even higher than what was reported for the heat-treated wrought Inconel 718 (355–385 HV). Moreover, for the thinner support wall thickness, an improvement in the hardness value of the fabricated samples was observed. This study urges a reconsideration of the common approach of selecting supports for additive manufacturing of samples when a higher quality of the as-fabricated parts is desired.

**Keywords:** laser powder bed fusion; Inconel 718; support structure

## 1. Introduction

The need for a stable, solution-strengthened, and non-hardenable alloy that can also operate at high temperatures (650 to 760 °C) has led to the development of nickel-based superalloy Inconel 718 (i.e., IN718) [1,2]. IN718 offers exceptionally high mechanical strength even at elevated temperatures (around 700 °C) [3], superior thermal resistance [4], and high resistance to corrosion and oxidation [5]. These beneficial features have made IN718 a great candidate in the sector of aircraft engines, turbine blades, combustion chambers, and nuclear reactors [6–8]. One major challenge associated with fabricating IN718, however, is the low machinability arising from its high hardness value (372 HV for wrought IN718 based on AMS 5663 and 350 HV for cast IN718 based on AMS 5383) and low thermal conductivity (11.2 W.m$^{-1}$.K$^{-1}$) [9], which, in turn, leads to extreme tool wear and unsatisfactory workpiece surface integrity [10,11]. In recent years, additive manufacturing technology has attracted great attention for the fabrication of IN718 since it minimizes the need for machining [12]. Amongst all available AM techniques, laser powder bed fusion (LPBF) has widely been adopted within the industry due to the overall cost and fabrication advantages [3,13–15].

During the past decade, significant work has been carried out in the context of improving the microstructure and mechanical properties of LPBF-fabricated IN718 through the change in LPBF process parameters [16–18]. In an initial attempt to increase the density and oxygen resistance of fabricated main parts, Jia et al. [19] investigated the effect of laser power ($P$) and scanning speed ($v$) on LPBF IN718. Their study revealed that both the oxidation resistance and density of the parts were improved with an increase in the $P/v$ ratio. In terms of the surface analysis, the role of laser processing parameters was found to be significant, too. Valdez et al. [16] reported the formation of discontinuity on the finished surface as well as the balling effect at a low laser energy power ($P$). On the other hand, at a relatively high value of the laser power ($P$), Parimi et al. [20] observed increased numbers of entrapped bubbles within the melt pools, thus a decreased level of density. Ravichander et al. [21] conducted a comprehensive study on the effect of laser processing parameters on the geometry accuracy of the as-fabricated sample. Out of all other energy density parameters (laser power ($P$), hatch spacing ($h$), and layer thickness ($t$)), they found laser scan speed ($v$) as a dominant factor towards grain length, which affects the final dimension of the part. Conducting microstructural analysis and mechanical testing on as-fabricated parts, Chlebus et al. [22] demonstrated the possibility of manufacturing IN718 parts with 99.8% density by double scanning strategy. However, the hardness value of their as-fabricated LPBF samples (maximum 322 HV) did not satisfy the minimum requirement for industrial applications (~355–385 HV [2]). Many research groups tried to enhance the microhardness value of the as-fabricated LBPF IN718 samples [14,19,23–36]. So far, Jia et al. [19] reported the highest achieved hardness value of as-fabricated LPBF IN718 samples in the literature as 395.8 HV, which was slightly above the minimum requirement per AMS5663. This encouraged more research to be conducted on post-process heat treatment of IN718 samples to push the microhardness to higher values.

Popovich et al. [37] reported the highest achieved hardness value of 478 HV for a post processed LPBF IN718 sample. The post process included hot isostatic pressing at 1180 °C under 21755.7 psi for 3 h and heat treated at 650 °C for 8 h. The heat-treated samples show clear distinct borders between microstructures of fine- and coarse-grained regions along with an increase in carbide density contributing to the hardness improvement. While the study proved that the achieved properties of IN718 are promising, post-process treatments are not ideal as they add to the overall time and cost of creating parts [38].

Support structures also play a significant role in the microstructure, composition, as well as the mechanical properties of LPBF-fabricated IN718. Jiang et al. [39] reviewed a total of 57 publications over the state-of-the-art research in the AM support structure area, but only found eight works [40–47] on the topic of support structure in metal LPBF. Low accessibility to metal 3D printers compared to polymer-based printers and higher expenses could be counted as the main reasons for such limited work on this topic. According to the literature, the role of adding support structures in LPBF falls into three different categories, which include cooling the part through heat conduction, reducing the risk of warpage during fabrication, and manufacturing cost reduction. They have been appropriately discussed by the articles in the literature as follows: (1) conducting the heat away from the part: the support structure can change the pattern of energy conduction from the molten pool to the building plate which can provide a stable thermal condition and thereby reduce the residual stresses within the sample [48]. For LPBF-fabricated parts, this effect has been investigated for the "overhang" state. Kajima et al. [49] studied the fatigue strengths of 45-degree overhanging arms additively manufactured with and without supports using Co-Cr-Mo powder. They observed that the fatigue strength was more than twice for the supported samples because of finer grains found in the microstructure as well as less defects and microcracks in the fracture surfaces of the supported samples. Moreover, lower residual strain was observed for the supported samples compared to unsupported ones. Chen et al. [50] performed thermal simulations on overhanging features when supported with powder versus a solid layer underneath. They reported that printing a part above a solid material would result in a lesser thermal gradient within the part, thus reducing the

resultant residual stress. They attributed this observation to the lower thermal conductivity of the AlSi10Mg powder, which was about 8 W/mK compared to that of around 90 W/mK for a solid phase of AlSi10Mg. However, the study was specific to a particular overhang design and did not consider microstructure variations at areas of the overhang supported by powder and that of as-fabricated parts. Moreover, post-processing operations can be time-consuming and more complex as the solid material should be removed from the main part. Therefore, minimizing the volume of the support structure plays an important role in the improvement of the efficiency of the fabrication process; (2) Preventing in-process failure: optimum support structures can reduce or even eliminate warping and distortion in LPBF-fabricated parts. Liu et al. [51] observed a reduced level of warping with a higher concentration of supports near the edges, where laser scanning starts and ends. According to Kruth et al. [52], however, the higher concentration of supports is associated with longer build time, along with difficulty in support removal, which might require further consideration. In a study conducted by Pinto et al. [53], it was revealed that a relatively low support concentration may lead to in-process failure due to the higher chance of particle clustering formation in the initial layers of the part, which they attribute to the increased magnetic interactions between particles. They also reported the same phenomenon for the reused powder materials and ferrite materials such as 316 L.SsteelPal et al. [54] attributed the formation of such defects to several physical actions as well as thermodynamic effects in the molten pool induced by laser and support structures. They suggested optimization of laser processing parameters, mainly the laser scanning speed ($v$), to reduce the formation of powder clustering in the initial layers of LPBF-processed Ti-6Al-4V. Such observations urge the need for support topology optimization, taking into consideration the factors like available area, loading, and constraint conditions [39]. (3) Reducing the overall production cost: The addition of support structures, compared to a solid support, decreases the overall fabrication cost because it is associated with a lower required amount of powder as well as post-process operations. In general, the main focus of the currently available support generator tools has been to decrease the amount of required supports and offer faster, cheaper, and more efficient AM processing [55,56]. These well-known and commercially available support generator software include Magics (Materialize, Michigan, USA), Sunata (Atlas 3D, Plymouth, IN), and 3DXpert (3DXpert, Rock Hill, SC). Generally, these packages offer several default support structure types for users with some control on the geometrical parameters of support structures (e.g., the wall thickness and gap). Due to the ease of use, these software packages have successfully been used by researchers in the LPBF design stage for generating support structures.

Overall the primary concern when designing support structures has always been to minimize the overall production cost and time, as well as lowering the risk factors associated with the fabrication, such as residual stress, the warping effect, and part collapse. However, no work has been conducted to enhance the quality of the as-fabricated main part through analyzing various factors including the microstructure behavior, material properties and fabrication cost. Inspired by this motivation, this study focused on the effect of support structure on the quality of parts considering the mentioned aspects. Specifically, the influence of support structure shape (i.e., Pin-type, Angled-type, Cone-type) and geometry (i.e., support wall thickness, and gap) was investigated on the composition, hardness, microstructure, and material/time consumption for LPBF-fabricated IN718 parts.

## 2. Materials and Methods

### 2.1. Computer-Aided Design

A total of seven main parts, each having a dimension of 8 mm × 8 mm × 4 mm, were modeled using Solidworks (version 2018–2019, Dassault Systems, Waltham, MA, USA). To study the influence of support type, three different support structures of Angled-type, Cone-type, and Pin-type were considered. The chosen support structure types were amongst the most widely used available options in the commercial support generation tools (i.e., 3D systems 3DXpert, Materialize Magics, Atlas3D Sunata, Siemens NX). For

an accurate comparison, a constant wall thickness of T = 0.53 mm and gap distance of G = 0.80 mm were considered (see Figure 1). Also, to evaluate the influence of the support geometry, four support structures of a particular type (i.e., angled-type) were designed by varying support wall thicknesses and gap distances in each of them. The angled-type support structure was chosen because it resembles the default support design selected by most of the aforementioned support generator software. This support structure is mostly used in software packages when the variation of the thickness is desired in support design. In all cases, the support structures had the overall dimension of 8 mm × 8 mm × 3 mm. Table 1 summarizes the variation in the type of support structures, the corresponding support wall thickness (T), and gap distance (G). For ease of referencing, the samples were labeled with a customized method. For example, in the label AT33G100, 'A' stands for the 'Angled-type support', 'T33' stands for the wall thickness of 0.33 mm, and 'G100' stands for the gap distance of 1.00 mm. The supports were allocated to the main part along with the appropriate labeling using the Materialize Magics software (Materialize, Plymouth, MI, USA).

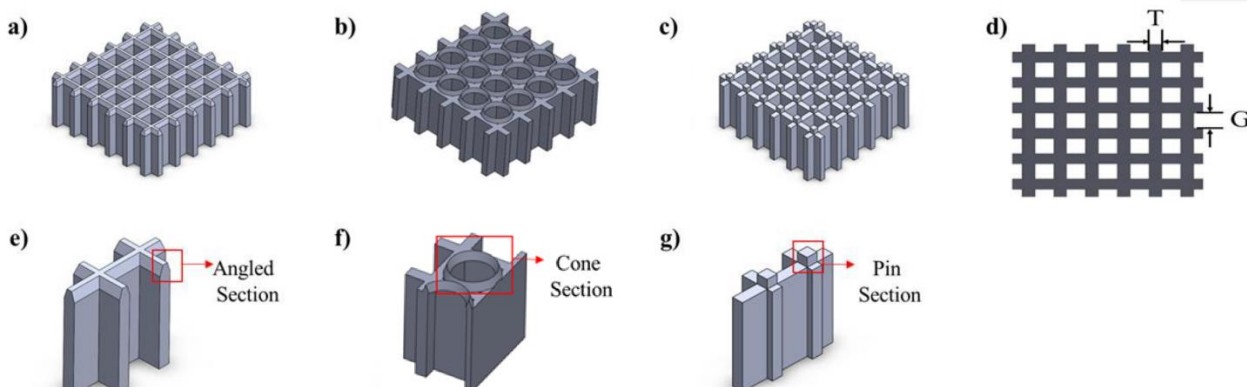

**Figure 1.** Computer-aided designs of the three different support structures: (**a**) angled-type (AT53G80), (**b**) cone-type (CT53G80), and (**c**) pin-type (PT53G80). (**d**) The thickness (T) and gap (G) of a support wall are presented from the top view. The cross-section view shows the contact area between the main sample and supports for (**e**) angled-type (AT53G80), (**f**) cone-type (CT53G80), and (**g**) pin-type (PT53G80).

### 2.2. Powder Preparation and Fabrication

IN718 powder was supplied from EOS GmbH (Krailling, Germany). The powder was sieved based on ASTM B214 [54] to avoid inhomogeneity in the distribution of particle size during fabrication. A scanning electron microscope (SEM) image of the fresh powder is presented in Figure 2a. The SEM analysis revealed that the powder had a spherical shape (the average circularity of 0.71), acceptable flowability and packing density, low impurity content, and excellent transformation ability. Moreover, the image was used for further analysis of the distribution of particle size using ImageJ software [57]. Figure 2b represents a histogram for the particle-size distributions of fresh IN718 powders.

An EOS M290 metal 3D printer (EOS GmbH Electro Optical Systems, Germany) equipped with a 400 W Ytterbium fiber laser was used to fabricate the designed parts with three repetitions. The laser processing parameter set used for the fabrication were laser power ($P$) of 285 W, scanning speed ($v$) of 960 mm/s, hatch spacing ($h$) of 110 μm, and layer thickness ($t$) of 40 μm, with an energy density ($E$) of 67 J/mm3, as calculated from the Equation (1) [58–60]. A stripe scanning strategy with a 67° rotation angle in each consecutive layer was employed for all the parts.

$$E = \frac{P}{h.v.t} \tag{1}$$

**Table 1.** Variation in the type and geometrical parameters of support structures. The abbreviation contained in the table has geometry type as the first letter followed by thickness and gap of the support geometry.

| Sample Number | Support Structure | Thickness (mm) | Gap (mm) | Support Label | CAD Design |
|---|---|---|---|---|---|
| Variation in the type of support structures | | | | | |
| 1 | Angled | 0.53 | 0.8 | AT53G80 | |
| 2 | Cone | 0.53 | 0.8 | CT53G80 | |
| 3 | Pin | 0.53 | 0.8 | PT53G80 | |
| Variation in the wall thickness of support structures (G = 0.8) | | | | | |
| 4 | Angled | 0.33 | 0.8 | AT33G80 | |
| 5 | Angled | 0.8 | 0.8 | AT80G80 | |
| Variation in the wall thickness of support structures (G = 1) | | | | | |
| 6 | Angled | 0.33 | 1 | AT33G100 | |
| 7 | Angled | 0.6 | 1 | AT60G100 | |

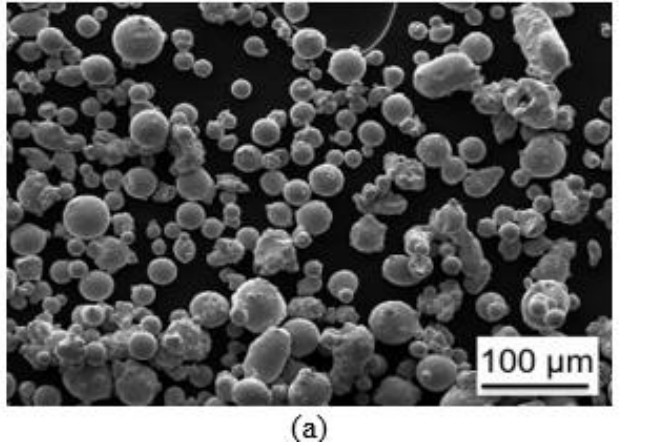

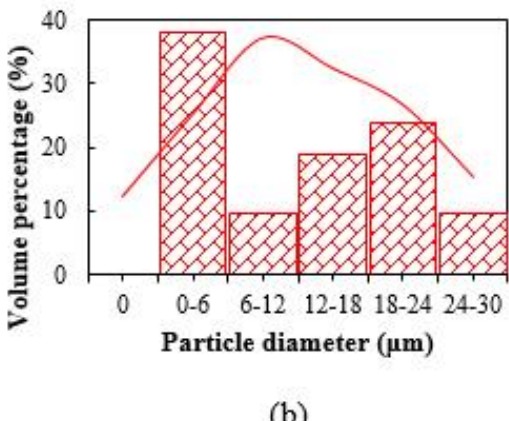

**Figure 2.** (**a**) SEM micrograph of the sieved IN718 powder; (**b**) particle size distribution of commercial EOS IN718 sieved powders.

### 2.3. Sample Preparation

The main samples were removed from the allotted support structures using the Wells No. 12 metal cutting bandsaw (Wellsaw, Three Rivers, MI, USA). An Allied Techcut 4 precision cutter (Allied High Tech, Compton, CA, USA) was then used to cut the bottom portion of the main samples 1 mm above the support/part interface through a plane normal to the building direction. The speed of the cutter blade was set at 100 RPM to obtain a

uniform cut. The schematic of the cutting strategy is presented in Figure 3. The polished surface and the surface used to perform XRD and the hardness analysis are as shown in Figure 3d.

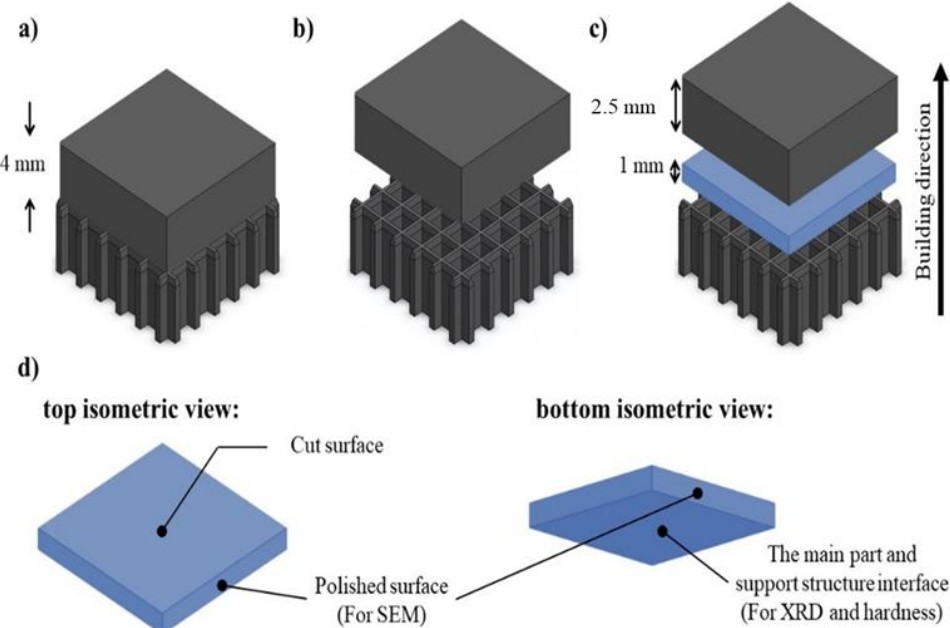

**Figure 3.** The schematic of (**a**) the fabricated main part on top of the allocated support structure, (**b**) the main part separated from the support structure, (**c**) the sectioning of the main part at 1 mm above the support/part interface through a plane normal to the building direction (0.5 mm reduction equal to the thickness of the cutter blade has been considered), and (**d**) the different views of the sectioned portion demonstrating the employed surfaces for the SEM and XRD characterization.

After this step, the side surface of the cut sample, as shown in Figure 3, was set in resin and polished using the Allied E-PREP 4™ Grinder/Polish machine (Allied High Tech, Compton, CA, USA). The samples were initially polished using sandpaper of grits 180 to 1200 with water as a lubricant. Next, the samples were polished with a 'DiaMat' polishing cloth with 1 μm polycrystalline diamond suspended solution, followed by 'Red Final C' polishing cloth with 0.05 μm colloidal silica solution. The samples and polisher components were cleaned with distilled water before every cycle and blown by compressed air to remove any debris. The samples were etched with Kalling's reagent before performing SEM analysis.

### 2.4. Experimental Procedure

The crystallographic analysis of the bottom surface (i.e., side facing the supports, as shown in Figure 3) of the as-built samples was determined using Bruker D8 Advance X-ray diffractometer (Bruker, Germany). The X-ray source was Copper (Cu) k-alpha, and the measurements were conducted at room temperature where the wavelength of X-rays was 1.5406 Å, step intervals of 0.02 and in 2θ between 35° and 100°. Vickers hardness analysis was performed with the help of a LECO LM 300 Vickers Hardness tester (LECO co., St. Joseph, MI, USA) on the bottom surface (i.e., side facing the supports, as shown in Figure 3) of the cut specimen. The test was conducted under 500 g loads applied for 10 s based on the ATSM E92-82 standard [61]. Five indentations were done for each sample, including the areas near and far from the edges at the interface plane (See Figure 3). It should be noted that the interface plane was selected for taking the measurements as this plane is affected directly by the support structures, without getting influence from the epitaxial growth occurring along the building direction. Moreover, consistency between XRD and hardness requires generating results at the same plane. In this case, the average

hardness value for each sample was reported for the interface plane. To evaluate the effect of support structures on the microstructure of the fabricated parts, a Hitachi S-3000N Variable Pressure SEM was used. To be able to investigate the effect of support structure on the depth of the melt pools, the side surface closer to the support zone (i.e., polished side surface, as shown in Figure 3) was selected for the SEM analysis and the dimensional analysis was performed according to the NASA MSFC-SPEC-3717 Standard [62]. It should be mentioned that, as the goal of study is evaluating the effect of support structure on the microstructure, the melt pool analysis was performed on the first few layers manufactured on top of the support structures (See Figure 3). Using the same equipment, energy dispersive X-ray spectroscopy (EDS) was performed on the samples to evaluate the compositional analysis. The microstructure features (i.e., melt pool), hardness, and composition of all the samples were evaluated and compared.

*2.5. Cost Model*

The total cost of fabricating a part was estimated by considering the direct raw material cost, the indirect cost, and the fixed cost [63]. Specifically, the direct raw material cost referred to the raw material purchase cost that was calculated based on the raw material market price $P_{raw\ material}$ (USD/g) and the weight of the part $w$ including both the main part and the support structure, calculated as follows.

$$w = \rho_{raw\ material} \times \left(V_{build} + V_{support}\right) \times \left(1 + r_{loss}\right) \tag{2}$$

$\rho_{raw\ material}$ is the density of the raw material (g/mm³), $V_{build}$ is the main part volume (mm³), $V_{support}$ is the support volume (mm³), and $r_{loss}$ is the material loss rate.

The indirect cost referred to all time-dependent costs that were associated with the capital investment and the fabrication process including the consumables like compressed air, Argon gas, and filters. The fixed cost $C_{fixed}$ was defined as all the fixed costs for each fabrication including machine setup and maintenance. The total cost was formulated as Equation (3).

$$C_{total} = \left(\dot{C}_{indirect} \times T_{build}\right) + C_{fixed} + w \times P_{raw\ material} \tag{3}$$

In this equation, $T_{build}$ denotes the total build time (hr) for fabricating the entire part and is estimated by using the machine control software EOSPRINT 2.0 in this research. $T_{build}$ is consisted of the time for fabricating the main part $T_{main\ part}$ and the time for fabricating the support structure ($T_{support}$). The $\dot{C}_{indirect}$ refers to the indirect cost rate (USD/hr) and it was calculated using the equation below.

$$\dot{C}_{indirect} = \dot{C}_{air} + \dot{C}_{Argon} + \dot{C}_{filter} + \dot{C}_{blade} + \left(C_{machine}/N_{machine} + C_{occupancy} + C_{Maintenance}\right)/M + \left(E_{build} \times P_{energy}\right) \tag{4}$$

In this equation, $\dot{C}_{air}$ is the cost rate of the compressed air (USD/hr), $\dot{C}_{Argon}$ is the cost rate of the Argon gas consumption (USD/hr), $\dot{C}_{filter}$ is the cost rate of filter usage (USD/hr), $\dot{C}_{blade}$ is the cost rate of the blade usage (USD/hr), $C_{machine}$ is the purchasing cost (USD), $N_{machine}$ is the lifetime of the machine (year), $C_{occupancy}$ is the occupancy cost per year (USD/year), $C_{Maintenance}$ is the maintenance cost per year (USD/year), $M$ is the maximum capacity of the machine (4,000 hrs/year), $E_{build}$ is total energy consumption per build (kW) and $P_{energy}$ is the average price of electricity (USD/kWh).

In addition, the fixed cost was calculated by considering the labor cost for machine setup and the cost of replacing and refinishing the build platform.

$$C_{fixed} = C_{setup} \times t_{setup} + C_{Platform}/N_{use} + C_{Refinishing} \tag{5}$$

$C_{setup}$ is the salary of the labor (USD/hr), $t_{setup}$ is the time required to setup the machine, $C_{Procument}$ is the build platform price (USD), $N_{use}$ is the total number of runs per

build platform, and $C_{Refinishing}$ is the platform refinishing cost. Parameters' values that are used in the cost calculation are shown in Table 2.

**Table 2.** List of parameters' values used in the cost calculation.

| Symbol | Definition | Values |
|---|---|---|
| | Indirect cost | |
| $\dot{C}_{air}$ | The cost rate of the compressed air | 0.00 (USD/hour) |
| $\dot{C}_{Argon}$ | The cost rate of Argon gas | 14.77 (USD/hour) |
| $\dot{C}_{filter}$ | The cost rate of the filters | 0.56 (USD/hour) |
| $\dot{C}_{blade}$ | The cost rate of the blade | 1.65 (USD/hour) |
| $C_{machine}$ | The capital investment of the AM machine | 125,941.71 (USD) |
| $N_{machine}$ | The useful life of the AM machine | 7 (year) |
| $C_{occupancy}$ | The occupancy cost rate of the AM machine | 3616.04 (USD/year) |
| $C_{Maintenance}$ | The yearly maintenance cost of the AM machine | 30769.23 (USD/year) |
| $M$ | The yearly utilization capability of the AM machine | 4000.00 (hour/year) |
| $E_{build}$ | The total energy consumption per build | 2.40 (kW) |
| $P_{energy}$ | The electricity price | 0.10 (USD/kWh) |
| | Fixed cost | |
| $C_{setup}$ | The labor hourly salary | 30.77 (USD/hour) |
| $t_{setup}$ | The machine setup time for each build | 3 (hour) |
| $C_{Platform}$ | The cost of the build platform | 275.00 (USD) |
| $N_{use}$ | The maximum number of uses of the build platform | 20 |
| $C_{Refinishing}$ | The refinishing cost of the build platform | 38.46 (USD) |
| $\rho_{raw\ material}$ | The density of the raw material | 8150.00 (g/mm3) |
| $r_{loss}$ | The material loss rate | 20.00% |
| | Direct raw material cost | |
| $P_{raw\ material}$ | The purchase cost of the raw material | 0.13 (USD/g) |

## 3. Results and Discussion

### 3.1. XRD and EDS Analysis

The XRD diffractograms of the as-built IN718 main sample revealed the presence of γ, γ′ and γ″ phases over the 2θ = 35–100 as shown in Figure 4. The δ and MC-type carbide phases were not detected in the XRD patterns as Seede et al. [64] showed the volume ratio of these phases in as-fabricated IN718 samples are minor and nondetectable with commercially available XRD instruments. As it is represented some of the peaks (γ, γ′, γ″ (220) and γ, γ′ (311)/ γ″ (033)) are asymmetric, which is also reported in the literature [65]. This observation can be attributed to two mechanisms: the dislocation distributions resulting in elastic strains [66,67] and variation in lattice parameters due to the compositional gradients in the microstructure of the samples [65]. However, more investigation is needed to explore this phenomenon, which is beyond the focus and goal of this study. In all the seven samples, the γ, γ′ (111)/γ″ (112) phases make up the dominant peak. Three phases of γ″ Ni3Nb with a D022 ordered body-centered tetragonal (bct) crystal structure, γ′ Ni3(Al, Ti) with a L12 ordered face-centered cubic (fcc) crystal structure and γ (Ni–Cr–Fe-C) in a face-centered cubic (fcc) crystal structure was detected for the main peak. The γ′ and γ″ are two main secondary phases known as the precipitation hardening phases of IN718 [68]. The next dominant peak was γ, γ′, γ″ (200) where overlapping of peaks exhibits the formation of precipitates parallel to the building direction which is one of the features of LPBF parts [35,64,69].

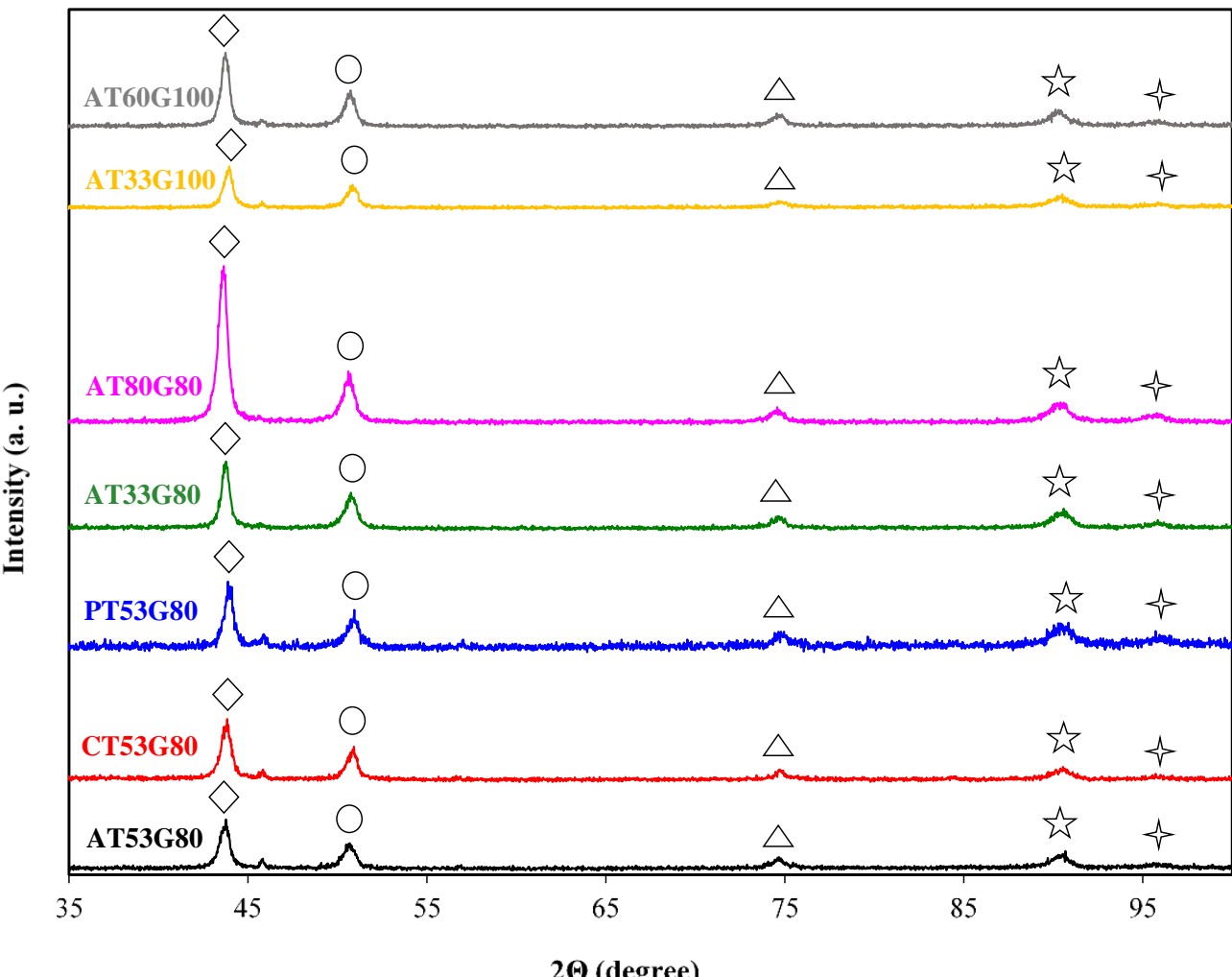

**Figure 4.** XRD diffractograms for the Inconel 718 main parts fabricated on top of different support structures. The $\gamma$, $\gamma'$ phases were coincident at (111), (200), (220). The symbols are defined as follows: $\gamma$, $\gamma'$(111)/ $\gamma''$(112): $\diamond$; $\gamma$, $\gamma'$, $\gamma''$ (200): $\bigcirc$; $\gamma$, $\gamma'$, $\gamma''$ (200): $\triangle$; $\gamma$, $\gamma'$(311)/ $\gamma''$(033): $\star$; and $\gamma$, $\gamma'$, $\gamma''$ (200): $\Upsilon$.

To have a more accurate characterization of $\gamma$, $\gamma'$ and $\gamma''$ phases, a smaller range of 2θ (42–45°) was represented in Figure 5. As it is shown, once the different type of the support structures was concerned, the Pin-type (PT53G80) presented a broader and shallower diffraction peak compared to the other two types (CT53G80 and AT53G80). Also, reduction in the thickness of the supports resulted in shallower peaks when the samples fabricated on top of AT60G100 and AT80G80 supports were compared with the samples supported by AT33G100 and AT33G80, respectively. A similar trend was observed when the gap parameter (G) increased, with a considerable drop in peak intensity being detected for the sample supported by AT33G100 compared to the sample fabricated with AT33G80 support.

In terms of the position of the peak angle, we can see that there is a variation for the $\gamma$ (111) diffracted angle between samples. The 2θ location of the detected $\gamma$ peaks is listed in Figure 6. When comparing the samples fabricated with different types of supports, the highest increase in the diffraction angle can be seen for the Pin-type support (PT53G80), followed by the cone-type (CT53G80). The geometrical parameters (i.e., thickness and gap) also affected the position of the diffraction angle (2θ). Reduction in the thickness of the supports (T) led to the increase in the diffraction angle, for both cases of AT80G80 and AT33G80, and AT60G100 and AT33G100. By comparing the AT33G100 and AT33G80, it was found out that a decrease in the gap value reduced the angle of diffraction.

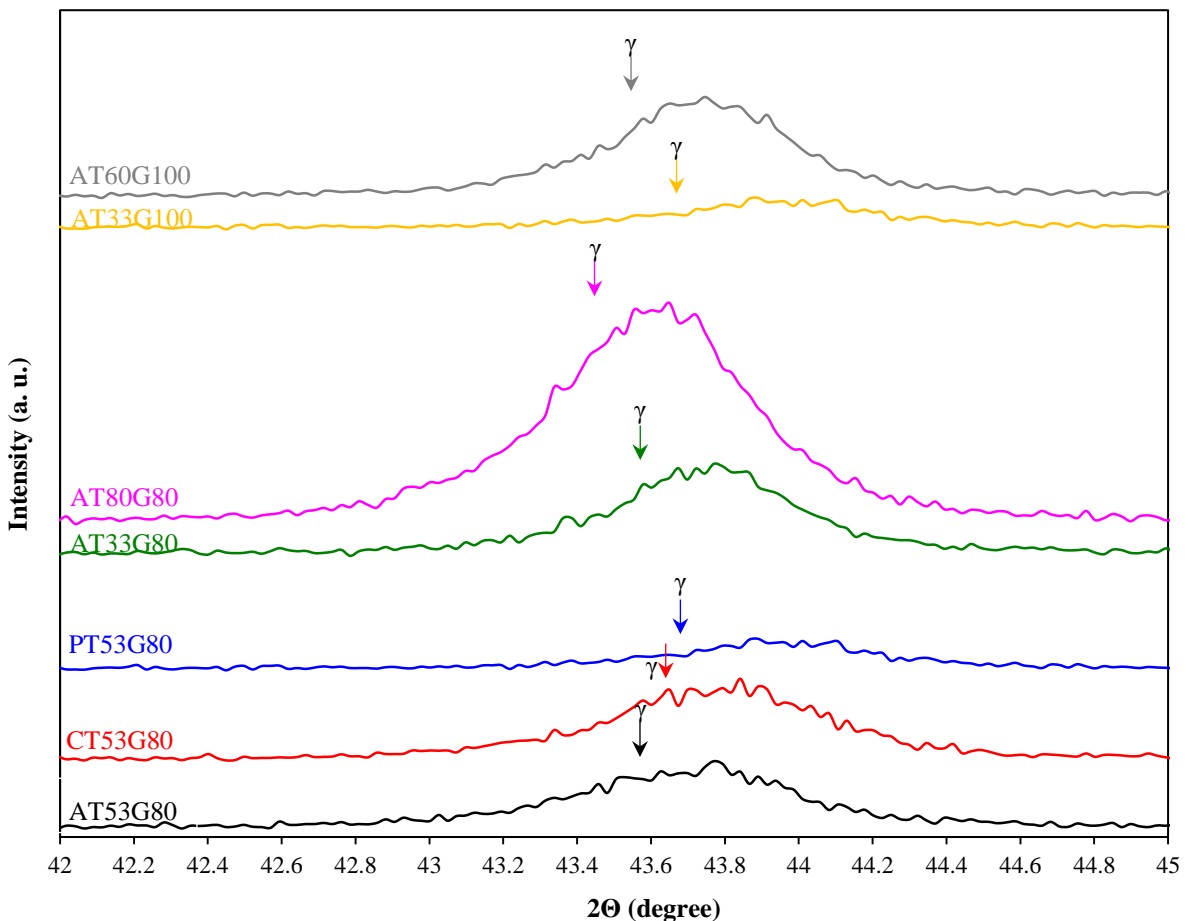

**Figure 5.** The variation in the γ (111) peak position in the XRD diffractograms extracted for the samples.

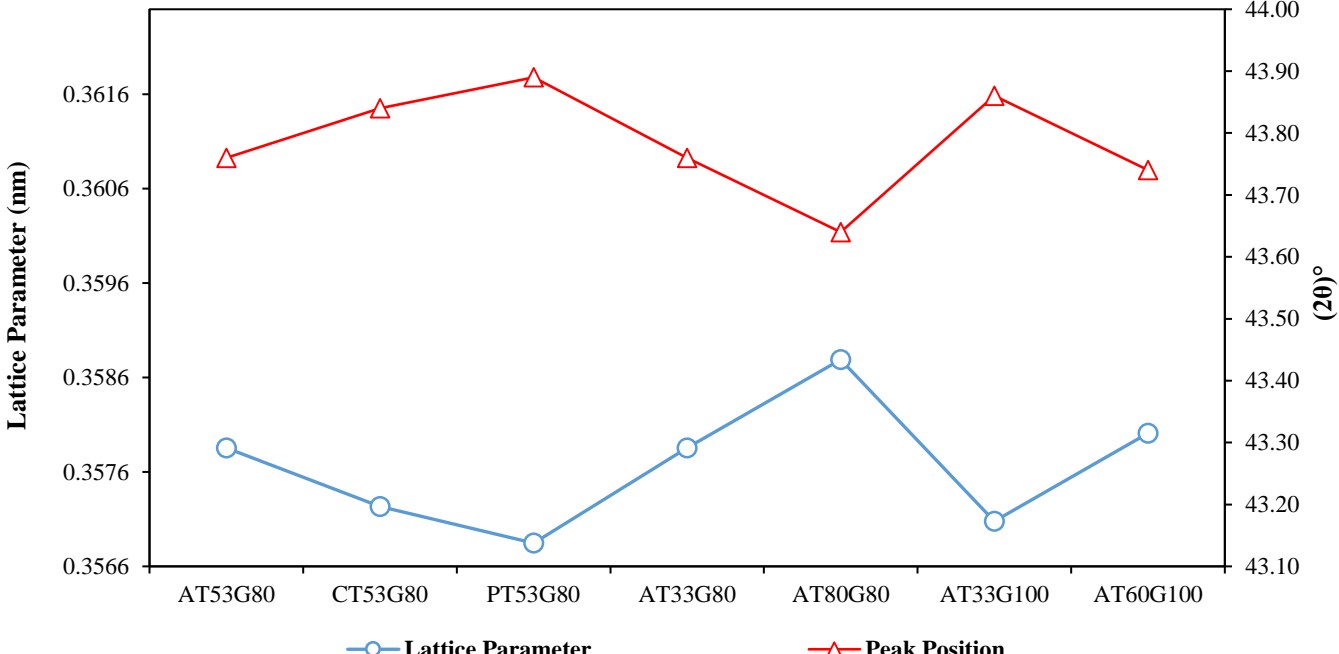

**Figure 6.** The displacement of identified γ (111) peaks and the lattice constant of the γ phase extracted from the XRD diffractograms of IN718 samples fabricated on top of different support structures.

Based on the variation observed for the diffraction angle, the lattice parameter can be calculated for each sample using the Bragg's law equation which relates the interplanar distance and the lattice parameter:

$$n\lambda = 2d\ sin\theta \tag{6}$$

where, $\lambda$ is the wavelength of the X-ray used in the XRD analysis, d is the interplanar distance and $\theta$ is the angle of diffraction.

Figure 6 shows the lattice parameter values determined for each sample. Among the samples fabricated with different types of supports, the highest lattice parameter (0.35822 nm) belongs to the samples fabricated with the Angled-type supports (AT53G80) and the lowest lattice parameter (0.35729 nm) belongs to the Pin-type supports (PT53G80). Also, in terms of geometrical parameters of the supports, the highest value of lattice size (0.3592 nm) belonged to the sample fabricated on the densest support with higher thickness and less gap value (AT80G80). Conversely, the sample supported by AT33G100 with the lowest thickness and highest gap, was formed by unit cells with smaller size. Therefore, as the dissolution of secondary phases into the γ phase leads to an increase in the size of the lattice parameter of the matrix phase, it can be inferred that the sample supported by the angled-type (AT53G80) experienced the higher level of dissolution of secondary phases, but the formation of less precipitates. By contrast, the lowest lattice size belonged to the sample with pin-type support which shows a higher level of precipitation of secondary phases. The same comparison can be done for the samples fabricated on angled-type support with different geometrical parameters. In this case, the sample fabricated on the support with a lower gap value (AT33G80) had a higher lattice size compared to the sample supported by the higher gap value (AT33G100). This brings about more dissolution but less precipitation for the former. When it comes to the effect of thickness, by comparing the samples supported by AT80G80 and AT33G80, higher thickness value resulted in higher lattice parameter size and in turn higher level of dissolution but less amount of precipitation of secondary phases. The same trend can be seen for the samples fabricated by AT60G100 and AT33G100 supports. This observation can be attributed to the heat transfer and cooling rate conditions vary among different types of support structures. It has been revealed that a higher cooling rate reduces the precipitate formation which leads to reduced precipitation hardening [70]. Since the area of contact between the support and the main part is directly proportional to the amount of heat conducted during the fabrication process, cooling rate and heat dissipation were affected by the type and shape of the support structure. To compare the level of energy dissipation via different support structure, the bottom and top area of the different supports which were in contact with the building plate and main part respectively are presented in Table 3. As the area of contact between the support structure and main part (top area) increased in order (PT53G80 < CT53G80 < AT53G80), the cooling rate of the supports would be increased according to the Fourier law of heat conduction [71]. The highest and lowest cooling rate can be estimated for the angled-type and pin-type supports owing to lesser and more precipitation on the samples fabricated with the respective supports. This can also be corroborated by the increasing volume of the supports with PT53G80 having the lowest volume and AT53G80 having the highest. As the volume of the support structure increases, more material will be present underneath to absorb the heat from a sintered layer of the part. Furthermore, AT80G80 with a higher thickness value and then higher contact area compared to AT33G80 resulted in the fabrication of samples with lower and higher amounts of precipitates, respectively. The same conclusion is true when AT33G100 and AT60G100 are compared. In terms of the effect of gap value, a higher level of precipitation can be expected for the AT33G100 with less contact area compared to AT33G80. It should be noted that this variation in the proportion of phases and precipitation levels can bring about different mechanical properties such as strength and ductility and then should be adjusted based on the desired application. Using the proper type of support structure, as observed here, can play a more important role to control the properties of the as-fabricated part.

**Table 3.** The top area, bottom area, and support volume for different support structures.

| Sample | Top Area (mm$^2$) | Bottom Area (mm$^2$) | Support Volume(mm$^3$) |
|---|---|---|---|
| AT53G80 | 28.18 | 40.77 | 120.46 |
| CT53G80 | 11.30 | 40.77 | 115.49 |
| PT53G80 | 10.08 | 40.77 | 113.11 |
| AT33G80 | 16.87 | 33.20 | 92.43 |
| AT80G80 | 38.69 | 48.00 | 142.63 |
| AT33G100 | 12.97 | 27.76 | 81.10 |
| AT60G100 | 28.00 | 39.00 | 115.11 |

Table 4 lists the EDS measured chemical compositions at the core of columnar dendrites that were obtained from identical samples with different support structures. As revealed from Table 4, the Ni element content has the highest value of 57.3 wt.% in the pin-type supported sample (PT53G80). This rate was dropped considerably for the angled-type support (AT53G80) with existing of the other elements being observed. Jia and Gu [19] found out that the Ni content increased by the increase in the energy input during the LPBF process of IN718 samples. A similar trend was observed in this study, as the highest Ni percentage can be seen for the sample fabricated on support with less contact area, which results in lower heat dissipation rate and thus longer heat accumulation (PT53G80). This condition resembled the fabrication of samples with a higher level of energy density. In terms of the effect of thickness, the same trend was observed by comparing the samples fabricated with AT33G80 and AT80G80 supports. A 5.5 percentage drop in Ni content was found for the sample fabricated on thicker support, which can be attributed to the larger contact area and higher cooling rate. However, the Ni content didn't change when the thickness increased from 0.33 mm in AT33G100 to 0.60 mm in AT60G100. Increasing the gap, however, increased the Ni content by more than 4 percent, which can be explained using the same logic discussed.

**Table 4.** EDS analysis showing chemical compositions at the core of columnar dendrites of LPBF-processed Inconel 718 parts fabricated using different support structures.

| Sample | Elements | | | |
|---|---|---|---|---|
| | Ni | Cr | Fe | Ti |
| AT53G80 | 46.1 | 17.6 | 16.1 | 1.0 |
| CT53G80 | 52.9 | 23.6 | 20.8 | 2.5 |
| PT53G80 | 57.3 | 21.1 | 20.3 | 1.2 |
| AT33G80 | 50.3 | 19.4 | 17.4 | 1.1 |
| AT80G80 | 44.8 | 16.3 | 14.2 | 1.2 |
| AT33G100 | 54.6 | 20.4 | 18.5 | 1.2 |
| AT60G100 | 54.5 | 20.6 | 18.5 | 1.2 |

### 3.2. Hardness Analysis

The mechanical properties of all the seven main parts were evaluated in terms of Vickers microhardness at room temperature, as represented in Figure 7. For comparison, the hardness value of the wrought IN718 [based on the AMS 5663, [72] and cast IN718 [based on the AMS 5383 [73]] are also presented in the same figure. To evaluate the effect of support type on the hardness, the angled-type of support (AT53G80), cone type support (CT53G80), and pin type support (PT53G80) were compared. From Figure 7, as the lattice parameter decreased, the hardness of the samples increased for the samples fabricated with different types of support. The PT53G80 sample resulted in the highest hardness of 460.5 HV, while the AT53G80 and CT53G80 support yielded a hardness value of 354.6 HV and 341.6 HV, respectively. This can be attributed to the different levels of secondary phases and precipitations explained in Section 3.1. It has been reported that the hardness of the IN718 alloy is dependent on the precipitation of two main secondary phases, $\gamma''$ and $\gamma'$

phases [74]. Another study conducted by Chang et al. demonstrated that the γ′ phase had a more significant role in the hardness of the as-fabricated IN718 [36]. In a similar study, Cao et al. [75] stated that a reduction in the strengthening phase leads to lower values of Vickers hardness. In agreement with the literature, the sample fabricated on top of the pin-type support structure (PT53G80) had a higher portion of secondary phases resulted in a higher hardness value. Moreover, the significant increase in hardness for this sample can be attributed to the crystal and grain size of the main part. As mentioned in Section 3.1, due to the broader and shallower diffraction peak observed for the sample with PT53G80 support, finer grain size can be inferred based on the Scherrer equation [76]. Also, based on the Hall–Petch equation, there is an inverse relationship between the hardness and grain size [77], which explains the highest level of hardness for the sample with finer grain size supported by PT53G80. However, since some other factors such as the condition of distribution of γ″ particles and the size of this strengthening phase can also affect the hardness of IN718 alloy, more investigation is needed to find the exact phenomenon that resulted in a significant rise in the hardness of this sample. In terms of geometrical parameters, a similar effect can be observed in the thickness variation for the supports. The sample fabricated with lower thickness support (AT33G80) showed a higher hardness value compared to the one fabricated with thicker support (AT80G80), due to the difference in level of precipitates discussed. The same trend was seen for samples with AT33G100 and AT60G100. However, by comparing samples supported by AT33G100 and AT33G80, it was observed that the gap value does not affect the hardness value significantly. It should be noted that, regardless of the sharp increase observed for the sample fabricated with pin-type support, using AT33G80 and AT33G100 supports resulted in higher hardness compared to value reported for the post-processed IN718 samples, regardless of the fabrication technique [74,78].

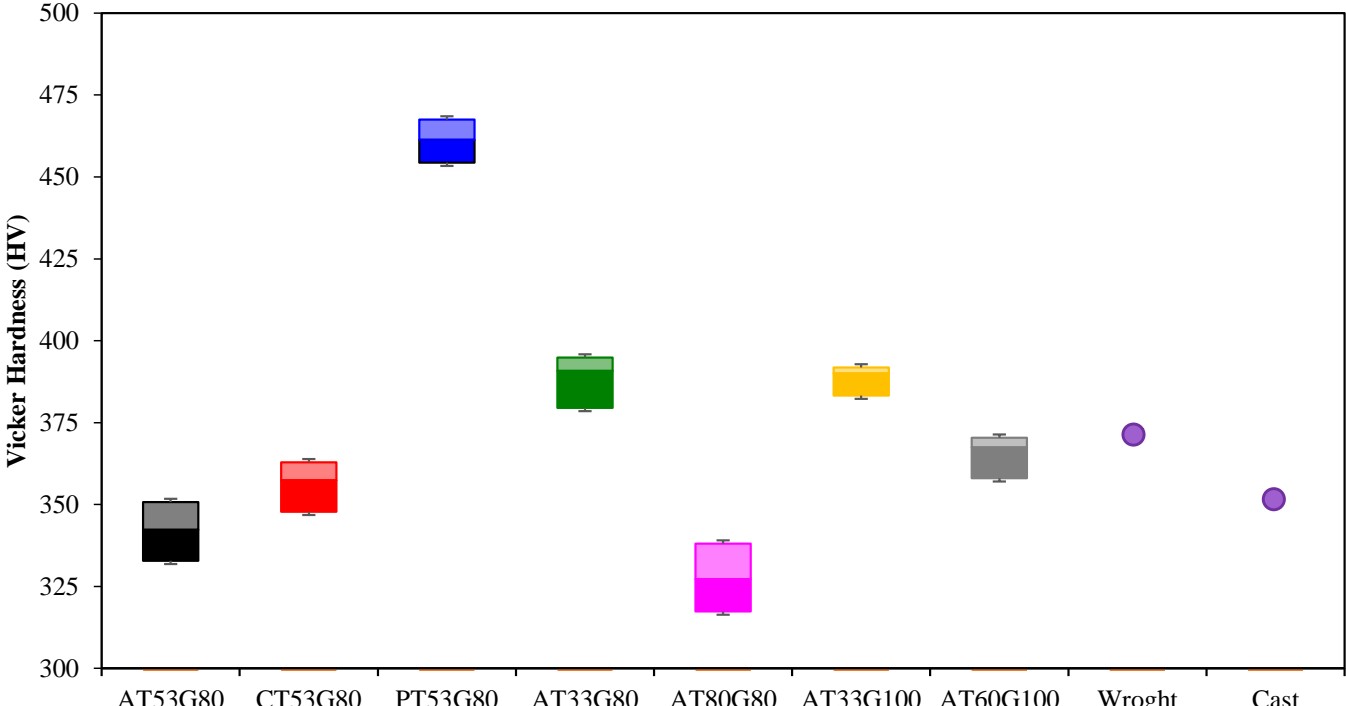

**Figure 7.** Vickers hardness evaluation of the LPBF-processed samples fabricated on top of support structures with different type and geometry. As reference, the Vickers hardness values of wrought IN718 (AMS 5663 [72]) and cast IN718 (AMS 5383 [73]) are also reported.

### 3.3. Microstructural Analysis

The SEM micrographs of the IN718 parts are presented in Figure 8. Figure 9 represents the results of the dimensional analysis performed on melt pool size for all the samples,

with the depth of melt pools being calculated. Regardless of the variation in size, the average depth of the pools was found higher than the initial layer thickness for all the samples. This proves the overlapping of laser tracks resulting in overlapping of melt pools, which has been reported as one of the thermal behaviors of the LPBF technique [79–81]. According to the results, the pin-type (PT53G80) gives rise to the highest average melt pool depth of ~79.1 μm followed by cone-type (CT53G80) to ~71 μm. However, the angled-type (AT53G80) showed the lowest average melt pool depth of ~62 μm in the fabricated sample. Therefore, as the contact area between the main part and the supports increased, shallower pools formed in the sample. This can be again attributed to the different thermal conditions and heat transfer rate experienced by the sample caused by different types of support used. A higher contact area between the main sample and support (see Table 3) facilitates heat dissipation through conduction and therefore faster solidification and less time for the molten material to penetrate into the deeper layers. This condition underwent by the sample with angled-type support causes the formation of shallower pools. Conversely, less contact area for the sample fabricated with pin-type support provides a better condition for flowing the molten material into underneath layers, re-melting these layers, and the overlapping of pools. This led to the formation of deeper pools in the microstructure of the sample. The observation is in agreement with previous studies where the formation of deeper pools was attributed to the numerous reheating cycles and laser overlapping [69]. It also has been reported that overlap between the melt pools associates with reheating cycles which acts similar to the heat treatment process. This causes more evenly dispersion of fine $\gamma''$ particles in the matrix [15,32,70] which improves the hardness value. This is consistent with the higher hardness value observed for the sample fabricated on pin-type support, with deeper pools being revealed in its microstructure. When it comes to the effect of geometrical parameters on the melt pool dimensions, the influence of thickness can be observed by comparing samples supported by AT80G80 and AT33G80, and also AT33G100 and AT60G100. A slight increase in depth of the pools can be seen for AT33G80 compared to AT80G80, and AT33G100 as against AT60G100. As expected, depth of the pools is higher for samples fabricated with a lower thickness value (i.e., less contact area). Therefore, it is consistent with the logic discussed for the different types of supports. Comparing the pool depth of the samples supported by AT33G80 and AT33G100 revealed that the gap value didn't affect the dimension of pools considerably.

### 3.4. Production Time and Cost Analyses

The actual production time for fabricating specimens with different support structures remained the same (32 min) in this research. The production time refers to the actual fabrication time excluding the required post-processing time, and it is calculated based on print speed and build path length. In this paper, the print speed selected for different specimens is the same; and the total length of the build path is similar. That's the main reasons why production time remains the same for specimens studied in this paper. In practice, specimens with more complex structures will require longer post-processing time, which is not considered in this research.

The estimated production costs for different specimens are slightly different for specimens studied in this research. The detailed cost calculation results are shown in Table 5. It can be observed from the table that all specimens have the same fixed cost and indirect cost, because each fabrication requires the same machine setup and maintenance, and the same consumption of consumables (such as compressed air, Argon gas, and filters). The direct material cost, on the other hand, is contributed by the material price and the consumption of materials of parts as well as the support structures; the direct material cost for different specimens remain similar because these specimens have the same weights of the parts and similar weights of the support structures. This means that the production cost of a specific specimen remains the same when different support structures are used but these structures have the same weight (assuming other factors remain the same such as the

values of process parameters), indicating promising opportunities to perform cost-aware support structure design.

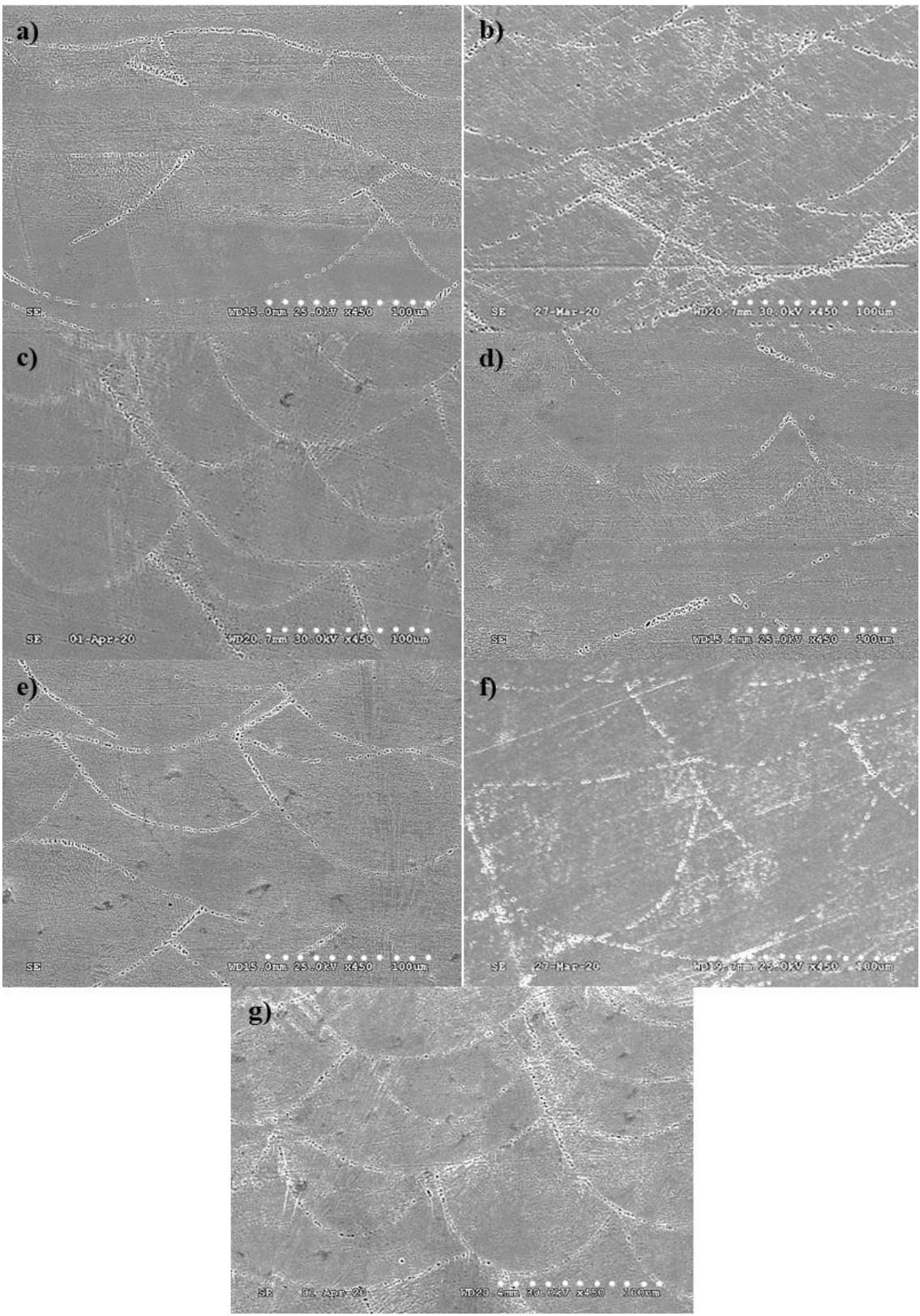

**Figure 8.** The SEM micrographs of the IN718 parts fabricated on top of different support structures: (**a**) AT53G80, (**b**) CT53G80, (**c**) PT53G80, (**d**) AT 33G80, (**e**) AT80G80, (**f**) AT33G100, and (**g**) AT60G100.

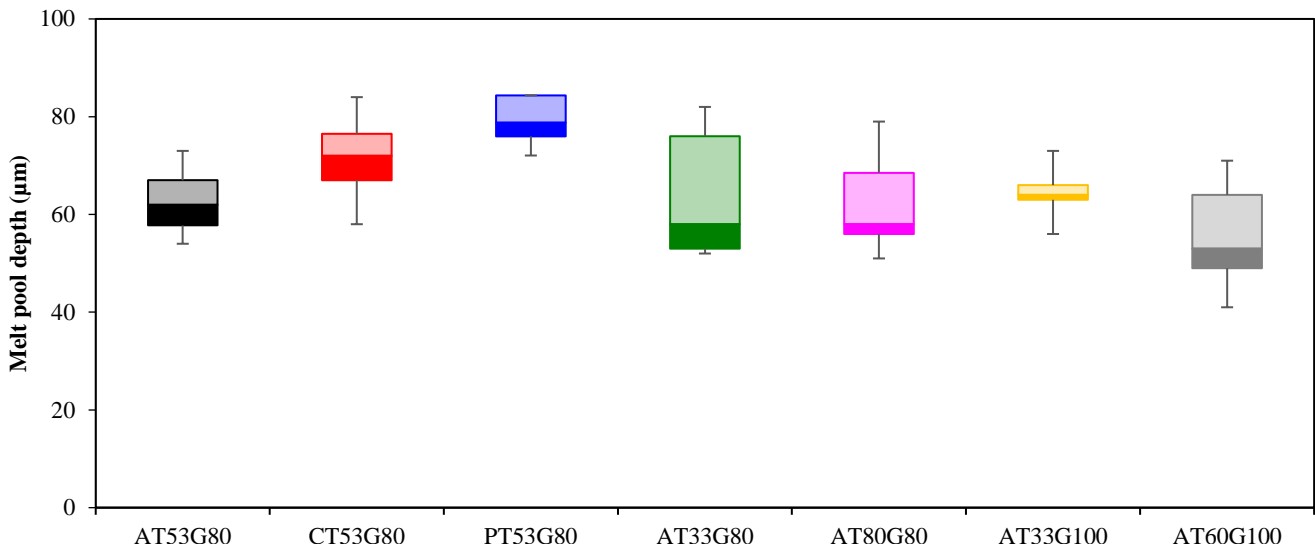

**Figure 9.** The melt pool depth of the IN718 LPBF parts fabricated on top of support structures with different types (AT53G80, CT53G80, PT53G80) and geometries (AT 33G80, AT80G80, AT33G100, AT60G100).

**Table 5.** The cost calculation results (cost: USD; volume: mm$^3$).

| Sample | Main Part Volume | Support Volume | Total Volume | Main Part Total Cost | Support Total Cost | Total Cost | Total Fixed Cost | Total Indirect Cost | Total Direct Cost |
|--------|------------------|----------------|--------------|----------------------|--------------------|------------|------------------|---------------------|-------------------|
| AT53G80 | 256.0 | 120.46 | 376.46 | 116.75 | 54.94 | 171.69 | 144.52 | 26.69 | 0.48 |
| CT53G80 | 256.0 | 115.49 | 371.49 | 118.31 | 53.37 | 171.68 | 144.52 | 26.69 | 0.47 |
| PT53G80 | 256.0 | 113.11 | 369.11 | 119.07 | 52.61 | 171.68 | 144.52 | 26.69 | 0.47 |
| AT33G80 | 256.0 | 92.43 | 348.43 | 126.12 | 45.53 | 171.65 | 144.52 | 26.69 | 0.44 |
| AT80G80 | 256.0 | 142.63 | 398.63 | 110.27 | 61.44 | 171.71 | 144.52 | 26.69 | 0.51 |
| AT33G100 | 256.0 | 81.10 | 337.10 | 130.24 | 41.29 | 171.63 | 144.52 | 26.69 | 0.43 |
| AT60G100 | 256.0 | 115.41 | 371.41 | 118.33 | 53.35 | 171.68 | 144.52 | 26.69 | 0.47 |

It can also be observed from Table 5 that different specimens have similar cost distributions among three cost components: fixed cost, indirect cost, and direct cost. As example of cost distribution among different cost components is shown in Figure 10 for specimen AT53G80. The cost calculation results in Figure 10 show that the fixed cost has the largest contribution at 84.17%, the direct cost is around 15.55% of the total cost, and the material-related direct cost is only less than 1% of the total cost. This indicates that in AM practices, despite the fact that raw material powders seem to be expensive, the material cost can be neglected compared to other consumables and machine setup and maintenance. In addition, the cost distributions among the main part and the support structure are shown in Figure 11. Among samples with different support structure types and geometry, their contribution to the total cost is similar, ranging from 24.06% (AT33G100) to 35.78% (AT80G80). The reason for this difference is mainly due to the support structure geometry (both thickness and gap). In addition, samples AT53G80, CT53G80, and PT53G80 demonstrate similar cost performance, indicating that the support structure type (angled, cone, and pin) does not evidently alter the cost contributions between the main part and the support structure.

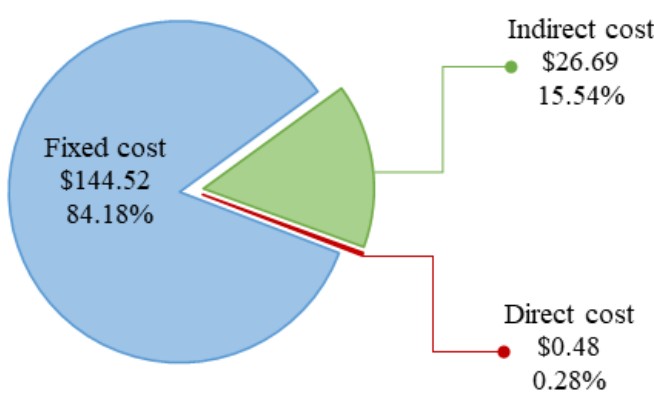

**Figure 10.** The cost distribution for specimen AT53G80.

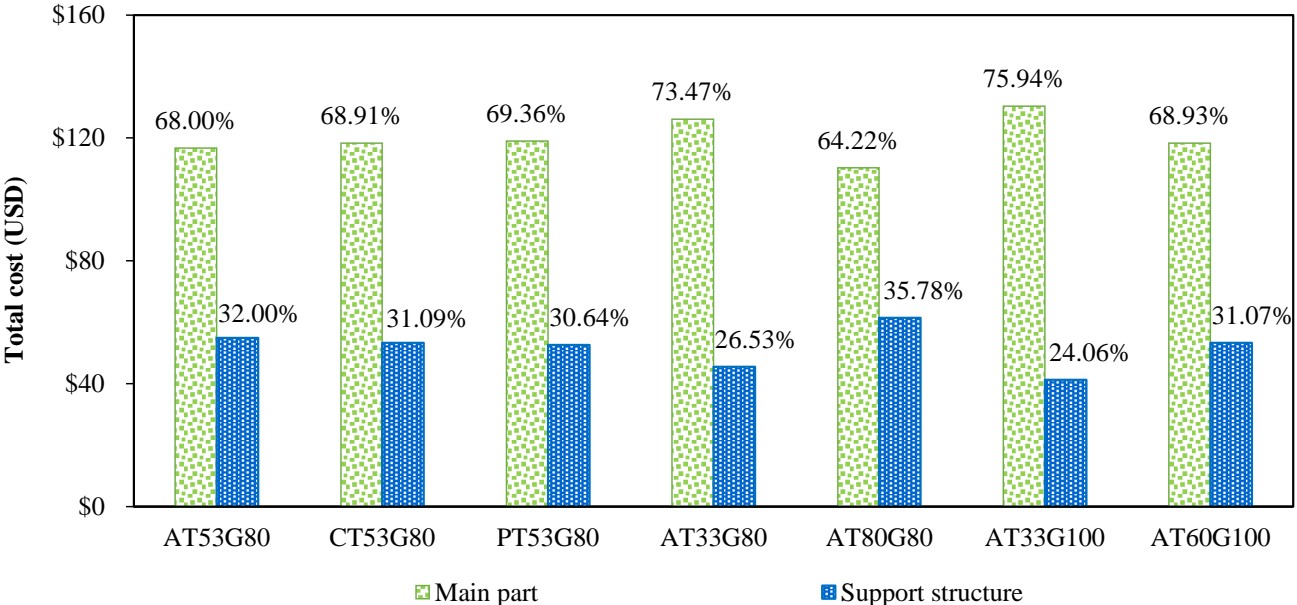

**Figure 11.** The total cost distribution between the main part and support structure for seven specimens.

Among seven samples, AT80G80 has the highest total cost of $171.71 and the highest total support cost of $61.44; AT80G80 demonstrates the lowest level of Vickers hardness (as shown in Figure 9). This indicates an interesting opportunity for improving the hardness level and reducing cost simultaneously. In addition, different types of the support structure (AT53G80, CT53G80, and PT53G80) show dramatically different levels of hardness (from around 330 to around 460, indicating almost 40% improvement) but they have similar total costs (from $171.68 to $171.69). In other words, adjusting the type of support structure could achieve better hardness "for free". Next, we focus on the same support structure but different geometries, i.e., different thicknesses (AT53G80, AT33G80, and AT80G80; AT33G100 and AT60G100) and different gap (AT33G80 and AT33G100). Samples with different thicknesses demonstrate evidently different levels of hardness and different total support costs while having similar total costs. For example, AT80G80 ($61.44) has around 35% more total support cost than AT33G80 ($45.54), and AT80G80 also demonstrates a lower hardness (around 325) than AT33G80 (around 380). On the other hand, samples with different gaps have a similar level of hardness and similar costs. AT33G80 and AT33G100 have similar averaged hardness (around 380) with different standard deviations, and similar total cost and total support cost.

## 4. Conclusions

The fabrication of IN718 samples through LPBF and the effect of support structures on the microstructure, composition, hardness, and fabrication cost of samples were studied. Seven identical samples supported by different types of support and different geometrical parameters were fabricated. Comparison between the properties and fabrication cost of samples revealed the following observations:

- Despite the presence of the same phases detected by XRD analysis in all the samples, the variation in the position of the diffracted peak angles and thus different lattice sizes were found for the specimens.

- Among the samples fabricated on different types of support, samples with pin-type (PT53G80) and angled-type (AT53G80) supports had the lowest and highest lattice size and therefore more and less level of precipitations, respectively. Also, in terms of geometrical parameters, the highest value of lattice size and the least precipitation level belonged to the sample fabricated on the densest support with higher thickness and less gap value (AT80G80). These observations were attributed to the heat transfer and cooling rate conditions among different types of support structures that arises from variation in the contact area between sample and support.

- It was revealed that the homogeneous or anisotropic behaviors of samples can be tailored using a proper type of support. In terms of the support type, the sample supported by pin-type (PT53G80) had the highest anisotropic behavior ratio among the samples. It was also found out that increasing the gap value as a geometrical parameter leads to more anisotropic behavior.

- The variation of the elements between samples was investigated using EDS Analysis. The lowest and highest Ni percentages were observed for the samples supported by Angled-type (AT53G80) and pin-type support (PT53G80), respectively. Also, increasing the gap increased the Ni content while no specific trend was observed for changing the thickness.

- Hardness of the samples varied among the samples fabricated with different types of support. A high microhardness value of 460.5 HV was achieved in the as-fabricated IN718 sample built on top of a pin support structure. The results were comparable with the LPBF IN718 in the literature for as-fabricated LPBF sample (322 HV) [21], heat-treated LPBF (335 HV), heat-treated plus hot isostatic pressing (478 HV) [37], and as-fabricated wrought and cast (353 HV) [37].

- In terms of geometrical parameters, the sample fabricated with lower thickness support (AT33G80) showed a higher hardness value compared to the one fabricated with thicker support (AT80G80), However, it was observed that the gap value doesn't affect the hardness value significantly. This variation in hardness value was mainly attributed to the different levels of secondary phases and precipitations between samples.

- Among the samples fabricated with different types of support, the deepest melt pools were observed for the pin-type support (PT53G80).

- Regarding the influence of thickness, a converse relationship was found between the depth of the pools and the thickness value. It was also found out that change in the gap value doesn't change the dimension of pools considerably.

- Despite the fact that raw material powder seems to be expensive, the actual material cost is less than other consumables as well as machine setup and maintenance.

- Adjusting the support shape and geometry have the potential of enhancing the specimen properties without adding extra cost. Also, it is possible to increase the hardness and reduce the cost simultaneously.

- Samples fabricated with the same support structure but different geometries (e.g., different thicknesses) demonstrate different levels of hardness and different total support costs while having similar total costs.

**Author Contributions:** Conceptualization, N.S.M., A.A. and Y.Y.; methodology, B.B.R., S.T., A.GR., B.F., M.H.; software, B.B.R., S.T., A.G.-R., B.F. and M.H.; validation, B.B.R., S.T., A.G.-R., B.F. and M.H.; formal analysis, B.B.R., S.T., A.G.-R., B.F. and M.H.; investigation, B.B.R., S.T., A.G.-R., B.F. and M.H.; resources, N.S.M., A.A. and Y.Y.; data curation; writing—original draft preparation, B.B.R., S.T., A.G.-R., B.F. and M.H.; writing—review and editing, N.S.M., A.A. and Y.Y.; visualization, N.S.M., A.A. and Y.Y.; supervision, N.S.M., A.A. and Y.Y.; project administration, N.S.M., A.A. and Y.Y.; funding acquisition, N.S.M. All authors have read and agreed to the published version of the manuscript.

**Funding:** This research was funded by University of Texas System STARs award.

**Institutional Review Board Statement:** Not applicable.

**Informed Consent Statement:** Not applicable.

**Data Availability Statement:** Not applicable.

**Acknowledgments:** This work was supported by a University of Texas System STARs award.

**Conflicts of Interest:** The authors declare no conflict of interest.

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
