# Peer review of "Cost-Aware Design and Fabrication of New Support Structures in Laser Powder Bed Fusion: Microstructure and Metallurgical Properties"

_applsci, doi:10.3390/app112110127_

Round 1

Reviewer 1 Report

Overall good work is presented by the authors and not much work is reported in this area. Some comments to improve the quality of the paper are as follows.

Line 18, IN718 is not defined in the abstract.

Line 29, “ operate at high temperatures (1200 to 1400 °F)” Please use degree Celsius for temperature throughout, as used in the next sentence.

Line 81, “According to what reported in the literature” please correct the sentence.

Line 82, “the role of adding support structures in LPBF falls into three different categories, which include: (1) …..” Please first highlight the three roles together within this sentence and then one by one explain each. At the moment the first role is at line 83, and the second is first explained at line 103; this affects the flow of the topic.

Line 112, “They also re-ported a higher risk of powder clustering formation and fabrication failure in ferrite ma-terials such as 316L steel, as well as reused powder materials.” Not clear, rephrasing is required.

Line 164, To make the work repeatable and more understandable, authors are requested to add the cross-sectioned views of the support structures as well in Figure 1. The best would be to add the draft views of each support type in Figure 1. Line 449 and line 455 where it is mentioned that pinned support has the lowest contact area and angled type has the highest contact area, this can only be visualized by the reader if there are draft views of the supports.

Also by looking at Figure 1, the naming rationale for the three supports structure is not clear, e.g., why authors call Figure 1(a) angled-type is not clear. Or these are just random names? If these are not random names, please provide more detailed images to clarify why these names are used for these supports.

Line 173 to 178. Figure 2 a and b are for fresh powder. Instead, these should be for the sieved powder or show both fresh and sieved powder images and distributions.

In section 2.3, the authors have not mentioned how they separated the main part from the support structure?

Figure 3, If the main part height is 4 mm, then after slicing it at 1 mm distance above the supports cannot leave 1 mm and 3 mm samples. The reduction will be equal to the thickness of the slicer blade thickness. Accordingly, correct Figure 3c.

Why the SEM and XRD are taken on different surfaces, explain it in the main text.

Line 221, “Five indentations were done for each 221 sample.” Where these readings were taken, in the middle or near the edges?

Why the hardness was not measured along the build direction (instead of just measured in one plane) so that the variation of hardness could be seen for different supports along the build direction.

Line 236 – 237. Superscripts are not used in units.

All sub-figures in Figure 8 are of poor quality (not well-focused images), and scales and text are barely readable. Also, the epitaxial grain growth is not apparent in Figure 8 contrary to what is claimed in Line365 and 366; instead only melt pools are visible. To provide clarity of your point, please show figures where grains can be seen clearly, add annotations within the figures, and provide build direction indication as well.

Line 411 to 425 are suitable for the introduction section but not for the discussion section.

Line 458 to Line 466, only results and trends are reported again like the previous section, and no underlying reasons are presented.

Line 467 to 490 is again just reporting the results, and then discussion and reasons are presented from line 491 onward.  This is unnecessarily increasing the length of the paper. Therefore, it is strongly recommended to combine the results and discussion sections to avoid repetitions.

Author Response

The authors collectively thank the reviewer for his/her insight and constructive comments and helpful feedback. The authors believe that addressing these comments has dramatically improved the quality of the manuscript. In the revised paper, the requested changes have been made as explained in this rebuttal. The authors hope that the reviewer finds our responses acceptable. All of the changes have been highlighted throughout the manuscript.

Here is a point-by-point response to the reviewer’s comments:

Overall good work is presented by the authors and not much work is reported in this area. Some comments to improve the quality of the paper are as follows.

Line 18, IN718 is not defined in the abstract.

Response:

Thank you for catching this error. It was defined in the abstract section as well. (Please see page 1, Line 10 - 11, Section: abstract)

Line 29, “ operate at high temperatures (1200 to 1400 °F)” Please use degree Celsius for temperature throughout, as used in the next sentence.

Response:

Thank you for your comment. The Fahrenheit degree was replaced by the proper SI unit (˚C) and the temperature values were updated accordingly. (Please see page 1, Line 29, Section: introduction)

Line 81, “According to what reported in the literature” please correct the sentence.

Response:

Thank you for this comment. The sentence was paraphrased using an appropriate phrase. (Please see page 2, Line 82, Section: introduction)

Line 82, “the role of adding support structures in LPBF falls into three different categories, which include: (1) …..” Please first highlight the three roles together within this sentence and then one by one explain each. At the moment the first role is at line 83, and the second is first explained at line 103; this affects the flow of the topic.

Response:

Thank you for your suggestion and we agree with your comment. As recommended by the reviewer, the main roles of adding supports were enumerated before discussing the details of each factor. (Please see page 2, Line 83 - 85, Section: introduction)

Line 112, “They also re-ported a higher risk of powder clustering formation and fabrication failure in ferrite ma-terials such as 316L steel, as well as reused powder materials.” Not clear, rephrasing is required.

Response:

Thank you for your suggestion. The authors paraphrased the sentence to avoid confusion. (Please see page 3, Line 115, Section: introduction)

Line 164, To make the work repeatable and more understandable, authors are requested to add the cross-sectioned views of the support structures as well in Figure 1. The best would be to add the draft views of each support type in Figure 1. Line 449 and line 455 where it is mentioned that pinned support has the lowest contact area and angled type has the highest contact area, this can only be visualized by the reader if there are draft views of the supports.

Response:

Thanks for your suggestion to present the structure of supports more clearly. According to the reviewer’s comment, figure 1 has been modified in the revised version. (Please see page 4, Line 167 Section: materials and methods)

Also by looking at Figure 1, the naming rationale for the three supports structure is not clear, e.g., why authors call Figure 1(a) angled-type is not clear. Or these are just random names? If these are not random names, please provide more detailed images to clarify why these names are used for these supports.

Response:

Thank you for your comment. The name for the supports was chosen based on their geometry and structural features. The reviewer is right as the figure doesn’t represent the structure of Angled-type clearly. Figure 1 was modified and more details were added to address the issue. (Please see page 4, Line 167, Section: materials and methods).

Line 173 to 178. Figures 2 a and b are for fresh powder. Instead, these should be for the sieved powder or show both fresh and sieved powder images and distributions.

Response:

Thank you for pointing this out. As mentioned in the text, the sieved powder was used for the fabrication and for the analysis as well. However, as noticed by the reviewer, the term “fresh powder” was used in the caption by mistake. The caption was corrected and the term “sieved powder” was used. (Please see page 5, Line 185, Section: materials and methods).

In section 2.3, the authors have not mentioned how they separated the main part from the support structure?

Response:

The authors thank the reviewer for pointing this out. The parts were separated form supports using a bandsaw. The details of the tool were added to the revised version. (Please see page 5, Line 194-195, Section: materials and methods).  

Figure 3, If the main part height is 4 mm, then after slicing it at 1 mm distance above the supports cannot leave 1 mm and 3 mm samples. The reduction will be equal to the thickness of the slicer blade thickness. Accordingly, correct Figure 3c.

Response:

The authors agree with the reviewer’s comment and they accordingly modified figure 3. (Please see page 6, Line 214-215, Section: materials and methods). 

Why the SEM and XRD are taken on different surfaces, explain it in the main text.

Response:

Thank you for your comment. In response to this comment, the authors have added an explanation regarding the selection of interest areas for the SEM results in the “Experimental procedure” section. (Please see page 6, Line 234 - 236, Section: materials and methods). 

Line 221, “Five indentations were done for each 221 sample.” Where these readings were taken, in the middle or near the edges?

Response:

Thank you for pointing this out. The measurements were taken on the horizontal plane including both the areas that are near and far from the edge. The appropriate explanation was included in the revised version. (Please see page 6, Line 227-228, Section: materials and methods). 

Why the hardness was not measured along the build direction (instead of just measured in one plane) so that the variation of hardness could be seen for different supports along the build direction.

Response:

Thank you for your comment. In order to see the direct effect of support structures on properties of the main part, the hardness measurements were taken on the horizontal plane at the interface between the part and supports. Also, the location is consistent with where the XRD results were generated from. In response to this comment, the authors have given an appropriate explanation. (Please see page 6, Line 228-232, Section: materials and methods). 

Line 236 – 237. Superscripts are not used in units.

Response:

Thanks for catching this error. The superscripts were used in units wherever it was required. (Please see page 7, Line 250-251, Section: materials and methods).  

All sub-figures in Figure 8 are of poor quality (not well-focused images), and scales and text are barely readable. Also, the epitaxial grain growth is not apparent in Figure 8 contrary to what is claimed in Line365 and 366; instead only melt pools are visible. To provide clarity of your point, please show figures where grains can be seen clearly, add annotations within the figures, and provide build direction indication as well.

Response:

The reviewer is right. The authors had mentioned the “epitaxial grain growth” as a common phenomenon for the laser powder bed fused parts which has been reported in the literature. However, as the focus of this study is not on the grain structure, this sentence was removed from the discussion. In addition, the quality of figure 8 was improved to present the microstructure of the melt pools more clearly. (Please see the changes as follows: page 15, Line 473, Section: Results and discussion)

Line 411 to 425 are suitable for the introduction section but not for the discussion section.

Response:

Thank you for your comment. The mentioned section was eliminated from the discussion section.

Line 458 to Line 466, only results and trends are reported again like the previous section, and no underlying reasons are presented.

Line 467 to 490 is again just reporting the results, and then discussion and reasons are presented from line 491 onward.  This is unnecessarily increasing the length of the paper. Therefore, it is strongly recommended to combine the results and discussion sections to avoid repetitions.

Response:

The authors thank the reviewer for this suggestion. As recommended by the reviewer, the discussion section was consolidated and combined with the results section. The recent version of the manuscript includes “Results and discussion” section with the content of the discussion being reduced by eliminating any parallels.

 Sincerely

Author Response

The authors collectively thank the reviewer for his/her insight and constructive comments and helpful feedbacks. Authors believe that addressing these comments has dramatically improved the quality of the manuscript. In the revised paper, the requested changes have been made as explained in this rebuttal. Authors hope that reviewer finds our responses acceptable. All of the changes have been highlighted throughout the manuscript.

Here is a point-by-point response to the reviewer’s comments:

  1. In the title, the authors use the terms “cost-aware design” and “new”. Moreover, in line 22, they write “this study urges a reconsideration of the common approach” and in line 135, they write “no work has been conducted to enhance the quality.” None of these statements applies to the current work. I agree that the authors present an effective way to calculate cost, but do not apply it to their design work and all the parts cost the same anyway. Moreover, the authors test common support structures found in numerous software, as they indicated, and yet I do not understand what differences they used that folks have not already tried before and currently do in industry. Moreover, there have been multiple studies conducted on improving quality of support structure, in which new companies have been spawned as a result of their own work, so I do not understand the third statement that no work has been done and that the current work enhances quality. The authors need to reread their manuscript and determine what they are trying to address so that rewriting will seem less presumptuous.

Response:

Thank you for your comment. Regarding your mention, “there have been multiple studies conducted on improving quality of support structure” we agree with that, but they neglected the effect of support on the quality of the main fabricated part, as their focus was only on the support structure. Conversely, the focus of this study is on microstructure and properties of the main part which is influenced by the support structure. This motivated the authors to conduct a more comprehensive study that looks into the provenance of properties of the main part influenced by using different support structures. The key answer to this question can only be found by evaluating the microstructure of the parts fabricated by different support structures, which to the knowledge of the authors has not been investigated in literature. Moreover, most of the previous studies investigated the effect of supports on the overhang structures to find the optimum overhang angle or the requirement of using supports. By contrast, the current study doesn’t limit the geometry of the parts to the overhang structures, instead simple cubic samples were selected for the analysis. In addition, it is common that users and manufacturing companies select the support structure design based on minimizing the material and fabrication cost. However, using the innovative cost analysis performed in the current study as a new aspect that has not been carried out before, it was revealed that the material and cost of the supports should be the least concern in selecting the right design. Instead, the quality and properties of the main part should also be considered as the most important factor.

Based on the recommendation of the reviewer and in order to avoid any confusion, the authors modified the goal of this study as follows: “However, no work has been conducted to enhance the quality of the as-fabricated main part through analyzing various factors including the microstructure behavior, material properties and fabrication cost. Inspired by this motivation, this study focused on the effect of support structure on the quality of parts considering the mentioned aspects.” (Please see page 3, Line 137-140, Section: Introduction)  

Overall, as discussed above, the goal of this study was investigating the variation of microstructure and properties of parts fabricated through different support structures. The goal of the study was not to optimize the geometry of support structures, but only focused on the commonly used options available in software packages. Along with the cost analysis results, the authors reported microstructural variation and in turn varying properties of the parts supported by commonly used structures. It is evident that there is no best or worst option for choosing the support structures. The reader can select the best support design according to the application of the component and desired properties.

  1. Table 1: Why not vary wall thickness for core and pin type structures. If you want to conduct a parametric design of experiments like this, it would make sense to test equally to all three support structures. Please provide a comment on this in the manuscript.

Response:

Thank you for pointing this out. Since Angled type supports are commonly used and their thickness and gap are allowed to be changed easily by the support generation packages, we varied them only for Angled type. This is mentioned in the manuscript as “The Angled-type support structure was chosen because it resembles the default support design selected by most of the aforementioned support generator software.” (Please see page 4, Line 156-158, Section: Materials and Methods). It should be noted that this is an initial study by the authors about the support structures and therefore the number of samples and variables were minimized to avoid complexity of the study. Optimization of the geometrical parameters for more support structure designs (especially pin-type support) will be the next step of this study, where authors can propose innovative designs for specific applications. According to the outcome of this study, the authors will consider more dimensional variables in the future studies. Therefore, variation of the geometry of supports through parametric design or optimization study will be a potential research topic for future but is not the focus of this current study. Therefore, as recommended by the reviewer this sentence was also added to explain the choice of the authors more clearly: “This support structure is mostly used in software packages where the variation of the thickness is desired in support design.” (Please see page 4, Line 158-159, Section: Materials and Methods)

  1. Line 213: XRD does not characterize composition, that is what EDS does. XRD is used to obtain crystallographic information, which can be used to determine phases present. Please remove the term “composition analysis” when referring to XRD. Line 227, please use the term “compositional analysis” when referring to EDS rather than “distribution of elements” since you are not mapping out the actual distribution of elements.

Response:

Thank you for this comment. The authors agree with this and accordingly they corrected the text as follows: “Crystallographic analysis” instead of “Composition analysis” (Please see page 6, Line 218, Section: Materials and Methods).

Also, the correct term was used for the EDS analysis by modifying the previous sentence as: “Using the same equipment, energy dispersive X-ray spectroscopy (EDS) was performed on the samples to evaluate the compositional analysis” (Please see page 7, Line 241-242, Section: Materials and Methods).

  1. Line 276: What is the “horizontal plane.” Are the authors considering the texture in the build direction?

Response:

Thank you for pointing this out. As depicted in the figure 3, the horizontal plane (perpendicular to the building direction) was selected for the XRD analysis. This horizontal plane was shown and labeled in the figure 3. As it acts as the interface plane between the part and the supports which was the area of interest in this study, the texture analysis was performed for this horizontal plane. This is also consistent with where the hardness measurements were performed. In order to provide more clarity for the reader, the following explanation was added to the manuscript: (Please see page 6, Line 228-231, Section: Materials and Methods) 

“It should be noted that the interface plane was selected for taking the measurements as this plane is affected directly by the support structures, without getting influence from the epitaxial growth occurring along the building direction. Moreover, consistency between XRD and hardness results requires generating results at the same plane.”

  1. The most interesting data is the XRD peak ratios. Supporting analysis will have to come from SEM or unetched areas to see the distribution of phases. There is no connection between the XRD peak ratios, Vickers hardness, and the melt pool analysis. Please only show data that provides meaning to the current work. In lines 353-359, it is difficult to make these comparisons with only two data points for Vickers hardness. Moreover, melt pools are difficult to measure in the bulk. Many researchers have dictated the best way to measure melt pools is by looking at the last layer melted. A straight horizontal line can be easily drawn through the data points in Figure 9 suggesting that really is no change in melt pool.

Response:

Thanks for your suggestion to present and discuss the results more efficiently.

The XRD peak ratios were provided to compare the level of homogeneity of the samples affected by different support structures. The authors found the results so interesting, as support structures show a significant effect on anisotropic properties of the main part. However, as the supporting results asked by the reviewer is out of focus of this study, this section was eliminated from the manuscript and will be used in the future studies. Regarding the Vickers hardness, not only two data points, but four data points (comparing 4 samples) have been selected to compare the results. The effect of thickness variation on the hardness values were compared for 2 cases to confirm the trend, one by fixing the gap as 80 mm (AT33G80 vs. AT80G80) and other by fixing the gap as 100 mm (AT33G100 vs. AT60G100). Since the trend on the hardness value for varying thickness matched for two gap cases, we concluded the influence of thickness on the hardness values.

When it comes to the melt pools, it should be mentioned that the dimensional analysis was performed according to the NASA MSFC-SPEC-3717 Standard. As reviewer mentioned, the melt pools formed in the last layer of the fabricated part should be considered for the dimensional analysis. However, since the goal of this study is to evaluate the effect of support structure on the microstructure, the melt pool analysis was performed on the first few layers manufactured on top of support structures (See figure 3). Hence, the authors used the same method mentioned in the standard, but for a region that is directly influenced by the support structures. It should be noted that the effect of support structures on the microstructure cannot be specifically determined by analyzing the last fabricated layer, as the epitaxial growth [1] and overlapping of melt pools [2] occurring between layers will influence the microstructure of the last layer and will complicate the analysis. The authors thank reviewer for this technical comment and will consider it for the future studies, where the indirect effect of support structures can be investigated throughout the height of the samples. In response to this comment, authors added appropriate reference and explanations to justify the method and area of the interest for dimensional analysis of the melt pools. (Please see pages 6, Line 234-240, Section: Materials and Methods). Moreover, according to the results (Figure 9), the reviewer is right that the average of the depth of the pools didn’t change significantly for most of the samples. However, as can be seen, using different type of support changed the average value around 30 percentage, which is not ignorable. The authors believe that this variation is high enough to present this result. Moreover, this variation in size of the pools (Specially for the Pin-type support) was confirmed by the different hardness values observed between the samples, as there is a relationship between size of the pools and the hardness value. The detailed discussion can be found in the manuscript in page 14, line 460-464, Results and Discussion section, Microstructural analysis.  

  1. The discussion is long winded. No one is going to read it. The results are merely repeated with an extra added line or two on cooling rate and possible distribution of phases. First of all, this is not a discussion and should be cut from the manuscript. The additional lines and interesting notes can be easily added to the results, making it a Results and Discussion section. Next, the authors found no real correlation between their results and postulate wildly on the possibility of differences in cooling rates and phase distribution. Phases can be quantified with XRD with methods such as Rietveld Refinement, giving some actual merit to the authors work. Moreover, there is no evidence on cooling rate changes and no data was provided on phase size or distribution. This reviewer strongly recommends that the discussion section is eliminated and what meaningful comments the authors have can be place in the Results.

Response: 

Thank you for your technical comment. Regarding the XRD analysis, as the reviewer is aware of the properties of the Inconel718 alloy, there are some challenges for preforming the Rietveld Refinement analysis. The authors have conducted a comprehensive literature review on this subject and found out that due to the peak overlapping between three phases of γ, γ' and γ", most of the studies in the literature didn’t perform quantification analysis to avoid reporting the inaccurate results [3-6]. There are also some studies worked on the quantification of the Inconel718 phases, but the accuracy of the results has not been evaluated. For instance, in the study by Popovich et al. [7], proportion of the γ phase has been reported to be 67.3 %. This extremely low value has not been reported in literature for the main γ phase and it needs more investigation. Similarly, using the Rietveld Refinement analysis performed on Inconel718, the proportion of the γ phase has been reported as 100 percent for the different areas of a sample [8]. This means that no precipitate is formed in the sample which for Inconel718 is less likely. The authors found an interesting method that has been used for deconvolution of the Inconel718 peaks [9]. The authors developed the same method and were able to extract the proportion of each phase. However, the authors need to evaluate the accuracy of the results generated by this method. The results are under further investigation through efforts in collaborating with other research groups. Therefore, authors decided to compare the crystallographic properties of the samples using peak position variation, which provides more accuracy.

Regarding the discussion, the authors agree with the reviewer’s comment. As recommended by the reviewer, the discussion section was consolidated and it was combined with the results section. The recent version of the manuscript includes “Results and Discussion” section with the content of discussion being reduced by eliminating any parallels.

  1. Lastly, the cost results come out of nowhere and in no way relate the other results of the paper. In lines 393-394, the others mention that the building time should be noted as the same. This should have been a major factor in the “cost-aware design” that was performed. Moreover, it turns out that all the parts cost the same. So, there is no real conclusion. I don’t see how this paper has helped anyone decide what supports to use.

Response:

The authors appreciate the reviewer for these insightful comments. We summarized the reviewer’s concerns in threefold and we would like to address these concerns point-by-point as follows.

(1) The observation that the print time and cost are similar for different specimens. First, the print time of different specimens is the same because the values of process parameters (i.e., laser power, layer thickness, travel speed, etc.) adopted to fabricate those specimens remain the same and the specimens’ overall height and the overall laser travel distance are similar. Second, the cost estimation for different specimens is similar (not exactly the same) because a) the fixed cost (cost fixed for each fabrication including machine setup and maintenance) remains the same for all specimens, b) indirect cost (time-dependent costs that are associated with the capital investment and the fabrication process including the consumables such as compressed air, Argon gas, and filters) remains the same for all specimens due to the equivalent print time, and c) the direct cost (material purchase cost) is different owing to different support structures and thus material consumptions. Although the direct material costs for those specimens are, as a matter of fact, different; their impact on the total cost per part rarely exists (less than 1%). The main reason is that the capital investment for the laser powder bed fusion machine used in this study is high, and this machine requires expensive machine setup and maintenance as well as consumables for each fabrication.

(2) The conclusions of the cost analysis in this study. The cost analysis presented in this study provides interesting observations that lead to the following conclusions. a) The production cost for a specific specimen remains the same when different support structures are used but these structures have the same weight, assuming other factors remain the same such as the values of process parameters, indicating promising opportunities to perform cost-aware support structure design. b) Different specimens have similar cost distributions among different cost components, indicating the most significant impact on total cost comes from the fixed cost and indirect cost; This means that in AM practices, despite the fact that AM raw materials seem to be expensive, the material cost can be neglected compared to other consumables and machine setup and maintenance especially in one-off productions. c) The observations from this study show that adjusting the type of support structure and the geometry of the support structure can potentially lead to better hardness “for free” or with no additional production cost

(3) How the results of this study can help AM designers and manufacturers. The results demonstrated in this research are helpful for AM communities when the design freedom on the support structure is presented. When selecting a type of support structure, and when designing the geometry of a support structure, there are many factors to consider: the fabricated microstructure, the achieved mechanical properties, the production cost of the main part and the support structure, etc. This research serves as an exploratory study and aims to investigate the interrelationships between these factors. Admittedly, this research has certain limitations such as limited types of support structures and the experimental nature of the study. We believe as more research in this area goes on, more comprehensive results will be generated that in the future, a quantitative tool to guide the AM design (of many aspects such as the main geometry, the support structure, the processes, etc.) will be developed upon all the exploratory studies we are doing now.

Sincerely

References:

  1. Cao, Y., et al., Grain growth in IN718 superalloy fabricated by laser additive manufacturing. Materials Science and Technology, 2020. 36(6): p. 765-769.
  2. Zhang, D., et al., Effect of standard heat treatment on the microstructure and mechanical properties of selective laser melting manufactured Inconel 718 superalloy. Materials Science and Engineering: A, 2015. 644: p. 32-40.
  3. Liu, P., et al., Microstructural Evolution and Phase Transformation on the XY Surface of Inconel 718 Ni-Based Alloys Fabricated by Selective Laser Melting under Different Heat Treatment. High Temperature Materials and Processes, 2019. 38(2019): p. 229-236.
  4. Amato, K., et al., Microstructures and mechanical behavior of Inconel 718 fabricated by selective laser melting. Acta Materialia, 2012. 60(5): p. 2229-2239.
  5. Jia, Q. and D. Gu, Selective laser melting additive manufacturing of Inconel 718 superalloy parts: Densification, microstructure and properties. Journal of Alloys and Compounds, 2014. 585: p. 713-721.
  6. Calandri, M., et al., Texture and microstructural features at different length scales in inconel 718 produced by selective laser melting. Materials, 2019. 12(8): p. 1293.
  7. Popovich, A.A., et al. Microstructure and mechanical properties of Inconel 718 produced by SLM and subsequent heat treatment. in Key Engineering Materials. 2015. Trans Tech Publ.
  8. Seede, R., et al., Microstructural and microhardness evolution from homogenization and hot isostatic pressing on selective laser melted Inconel 718: structure, texture, and phases. Journal of Manufacturing and Materials Processing, 2018. 2(2): p. 30.
  9. Li, X., et al., Effect of heat treatment on microstructure evolution of Inconel 718 alloy fabricated by selective laser melting. Journal of Alloys and Compounds, 2018. 764: p. 639-649.

Round 2

Reviewer 2 Report

The authors have adequately answered my questions and concerns. I appreciate the merging of the results and discussion and think that will better suit the authors.